# PERCEPT myeloma: a protocol for a pilot randomised controlled trial of exercise prehabilitation before and during autologous stem cell transplantation in patients with multiple myeloma

Orla McCourt  ,[1,2] Abigail Fisher,[3] Gita Ramdharry,[4] Anna L Roberts,[3] Joanne Land,[3] Neil Rabin,[5] Kwee Yong[2]

For numbered affiliations see end of article.

**Correspondence to**
Ms Orla McCourt;
o.mccourt@nhs.net

## ABSTRACT

**Introduction** Myeloma, a blood cancer originating from plasma cells, is the most common indication for autologous stem cell transplantation (SCT). Patients with myeloma undergoing autologous SCT (ASCT) experience treatment-related morbidity and reduction in function and well-being for many months post-treatment. Interventions targeting physical functioning delivered prior to and during SCT have shown promising results in mixed haematological populations and may offer a non-pharmacological solution to physically optimising and preparing patients for SCT. The aim of this study is to investigate the feasibility of a physiotherapist-led exercise intervention as an integral part of the myeloma ASCT pathway at a UK tertiary centre.

**Methods and analysis** PERCEPT is a single-site, pilot randomised controlled trial of an exercise intervention embedded within the myeloma ASCT pathway, compared with usual care. The primary study end points will be feasibility measures of study and intervention delivery including recruitment rates, acceptability of intervention, study completion rate and any adverse events. Secondary end points will evaluate differences between the exercise intervention group and the usual care control group in cancer-related fatigue, quality of life, functional capacity (6 min walk test; handheld dynamometry; a timed sit-to-stand test) and objective and self-reported physical activity. Outcomes will be assessed at four time points, approximately 6–8 weeks prior to SCT, on/around day of SCT, on discharge from SCT hospital admission and 12 weeks post-discharge. The exercise intervention comprises of partly supervised physiotherapist-led aerobic and resistance exercise including behaviour change techniques to promote change in exercise behaviour. The primary outcomes from the trial will be summarised as percentages or mean values with 95% CIs. Group differences for secondary outcomes at each time point will be analysed using appropriate statistical models.

**Ethics and dissemination** This study has NHS REC approval (Camden and Kings Cross, 19/LO/0204). Results will be disseminated through publication and

### Strengths and limitations of this study

► This study features a pragmatic trial design integrated into the existing autologous transplant treatment pathway and the use of a partly supervised exercise intervention, which will incorporate behaviour change techniques (BCTs), in myeloma patients before, during and after stem cell transplantation.
► It is the first study of a prehabilitation and rehabilitation exercise intervention in myeloma transplant recipients in the UK to incorporate a control arm.
► Although the intervention includes exercise and incorporated BCTs, it does not include nutritional assessment and/or dietary intervention, as recommended in guidance for prehabilitation interventions.
► This study is a single-site pilot trial and is not statistically powered.

presentations at haematology and rehabilitation-related meetings.

**Trial registration number** ISRCTN15875290.

## INTRODUCTION

Myeloma, also known as multiple myeloma, is an incurable blood cancer of plasma cells in the bone marrow and accounts for 10% of all haematological cancers.[1 2] The incidence is approximately 22 cases per 100 000 people in the UK.[1] Although incurable, improved understanding of disease mechanisms and advances in treatments means that survival in myeloma is increasing at the fastest rate among all cancer types.[3] Median-estimated survival time for myeloma has quadrupled over the last four decades and 5-year survival is now around 47%.[4]

Myeloma patients are treated with autologous stem cell transplantation (SCT) if eligible

(based on age and performance status).[5] Autologous SCT (ASCT) involves harvesting the individual's own stem cells and re-infusing them after high-dose chemotherapy.[6] ASCT has been shown in multiple randomised controlled trials (RCTs) to prolong progression-free survival, and in some studies overall survival in myeloma.[2 7] In the UK, the total numbers of ASCTs carried out each year have been increasing by 5% each year.[8] Data from the British Society of Blood and Marrow Transplant register shows that myeloma is the most common indication for ASCT in the UK, with 1411 procedures taking place in 2016.[9] However ASCT has a number of side effects,[5] including repeated infections, nausea, anorexia and fatigue and patients can suffer a reduction in their functionality and well-being for 6 to 12 months post-transplantation.[10] There is a need to develop supportive interventions for patients undergoing ASCT aimed at optimising patient fitness before and during ASCT in order to mitigate or minimise the negative effects of treatment on physical and psychological outcomes.[11] Prehabilitation and rehabilitation, targeting physical function and fitness, have been suggested as an integral component of myeloma treatment pathways in guidance for managing the consequences of the disease and its treatment; however, further research is required to better understand the potential impact of prehabilitation/rehabilitation before and after treatment for myeloma.[12] Given the prevalence of myeloma-related bone destruction in this population, it is also vital to develop and test tailored exercise interventions that are safe and effective at enhancing physical capacity in the presence of disease-related skeletal deformity and pain, common symptoms that limit patients with myeloma being physically active during and after treatment.[13]

There is emerging evidence that exercise during or after SCT may be beneficial. However, existing studies have generally been small, with methodological limitations, particularly the heterogeneity of the interventions and mixed haematological cancer populations studied.[14 15] Two systematic reviews, of 8 and 11 RCTs, respectively, have shown that exercise is safe and feasible for people being treated for different types of blood cancer during hospital admission and that the best results are seen when exercise is introduced before or during treatment, when compared with usual care. The majority of the studies included in these reviews were carried out with patients undergoing allogeneic SCT or mixed samples of both allogeneic SCT and ASCT recipients using varied interventions.[16 17] There is acknowledged variation in symptoms and consequences of treatment experienced by patients with different bloods cancers, and between undergoing allogeneic SCT and ASCT.[16 18] Studies among allogeneic SCT recipients highlight the intensive nature of the medical intervention and its associated symptom burden and consequences, most notably graft versus host disease (GvHD) and an increased risk of treatment-associated mortality.[19] In contrast, with autologous transplantation the mortality is considerably lower and there is no risk of GvHD, with its attendant risk of infections and short-term and long-term morbidity.[18] These variations in clinical features support the need to study rehabilitation interventions in uniform blood cancer patient populations.

Only two small trials have explored the use of a pre-transplant exercise in a myeloma only population undergoing ASCT, with mixed results.[20–22] In the first study, a feasibility study of 24 participants showed a significant increase in lean muscle mass in the exercise group.[20] The second study of 166 patients reported a decline in aerobic capacity and increased fatigue in both the experimental and control group, with no difference between groups after SCT. This study, reported in two papers, was also testing the effect of prophylactic epotin alfa therapy, a medical product that raises haemoglobin levels, alongside exercise and was therefore not solely assessing the efficacy of the prescribed exercise.[21 22] Additional limitations to these studies include provision of an unsupervised home-based exercise intervention and that the control groups were also advised to walk 20 min/ day,[20 21] which may have diluted the effect size. These studies, however, have demonstrated that it is safe for myeloma patients to exercise while undergoing ASCT.

Limited published literature exists regarding physical activity (PA) or exercise interventions in patients undergoing SCT originating in the UK. A single-arm pilot study of a physiotherapist-led, tailored, exercise intervention delivered to myeloma survivors post-treatment conducted at the authors' centre demonstrated that exercise was associated with improvements in quality of life (QOL), cancer-related fatigue and muscle strength, and it was extremely well received by patients.[23] A single-arm study to test feasibility of delivering exercise during ASCT among people with myeloma in a centre in Sheffield has completed recruitment but the results are yet to be published.[24]

## Objectives

This primary aim of this pilot, single-centre, RCT is to investigate the feasibility of a physiotherapist-led exercise intervention as an integral part of the ASCT pathway for patients with myeloma. It is hypothesised that this intervention, delivered before and during ASCT treatment for myeloma, will be feasible in terms of recruitment rate and willingness of the participants to be randomised, intervention adherence, compliance to the exercise prescription and attrition due to the intervention.

Primary objectives of the pilot study are to measure eligibility and recruitment rates from potential participants screened on referral to myeloma outpatient clinics for consideration for ASCT; acceptability of the intervention, including adherence to exercise sessions and deviations from exercise protocol; study completion rate and any adverse events.

Secondary objectives are to collect preliminary data on the intended primary/secondary outcomes of a future fully powered RCT to obtain mean and SD estimates to inform a sample size calculation. These outcomes include: cancer-related fatigue; QOL; levels of PA and

sedentary time; functional capacity and muscle strength; hospital length of stay, rates of readmission and health service utilisation and if the trial/exercise intervention is delivered as intended.

Exploratory objectives are to investigate any effect on immune parameters (including immunoglobulins, T cells, B cells, natural killer cells).

## METHODS AND ANALYSIS
### Participants
All patients with a diagnosis of myeloma referred to University College London Hospitals NHS Foundation Trust (UCLH) for consideration for ASCT will be identified through discussion with the clinical multidisciplinary team (MDT) responsible for coordinating treatment for patients with myeloma, and the team includes the medical team and clinical nurse specialists (CNSs), nurses who act in a key working role to provide support for patients undergoing cancer treatment.[25]

### Inclusion and exclusion criteria
Potential participants will be screened using the inclusion/exclusion criteria by the research team and through discussion with the haematology consultants, CNSs and MDT. All adult patients with myeloma referred to UCLH for consideration of ASCT, who are clinically able to carry out an exercise training programme on a regular basis and willing and able to provide written informed consent will be deemed eligible to approach. Patients must have a good command of written and spoken English.

Patients will be excluded if they have known spinal instability, spinal cord compression or neurological deficits, have had recent (within 6 weeks) spinal surgery or other surgery for pathological fractures, have an abnormal resting ECG and/or unstable angina. Patients will also be excluded if they are unable or unwilling to undertake an exercise programme on a regular basis or are unable or unwilling to provide informed consent.

The initial aims of this study were developed with input from participants in myeloma survivorship research conducted at our centre. The study has been further developed through involvement of the UCLH Cancer Patient and Public Advisory Group (CPPAG) during the preparation of the grant application and development of the study protocol. The CPPAG will continue to be provided with updates on the progress of the study and its results.

### Setting
UCLH is an acute NHS trust in London, England and tertiary referral centre for SCT. Recruitment and outpatient assessments will take place at the University College Hospital (UCH) Macmillan Cancer Centre. Weekly visits to attend the pre-transplant intervention exercise will take place at UCLH in a gym setting. Exercise and assessments during admission for autologous transplant will take place either in the UCH Macmillan Cancer Centre or on the haematology inpatient wards at UCH.

### Recruitment procedures
A participant information sheet (PIS) detailing the study will be provided in a clinical information pack routinely posted to patients prior to their first routine transplant clinic appointment. Potential participants will be contacted by the principle researcher or study physiotherapist, with a follow-up telephone call to allow patients to ask questions about the study. Those who wish to proceed will be asked to provide informed consent and this will be recorded on the participant consent form. Those approached to take part in the study who are subsequently found to be ineligible or decline to take part will be recorded in a screening log by the principle investigator.

It is conservatively estimated that approximately 60–75 patients will be recruited over a 15–18-month period, with completion of final follow-ups within 18–21 months. Approximately 100 myeloma patients undergo ASCT at UCLH each year.

### Randomisation procedures and blinding
Participant randomisation will be undertaken centrally by a researcher not involved in outcome assessment or intervention delivery, using minimisation, with age and gender as the stratification factors using MinimPy, a free, open-source, desktop programme (https://sourceforge.net/projects/minimpy/). This researcher will communicate their allocation of participants to the physiotherapist who will inform participants of the allocation and deliver the intervention to those in the intervention arm. The principle investigator will attempt to remain blinded to group allocation and will conduct follow-up outcome assessments.

### Intervention
The intervention will involve a partly supervised exercise intervention, incorporating behaviour change techniques (BCTs), delivered by a physiotherapist and individually tailored to the ability of each participant. The intervention will involve aerobic and resistance exercise during three phases of the ASCT pathway:
► Phase 1 (6–8 weeks before hospital admission for ASCT): Participants will receive one exercise session per week, supervised by a physiotherapist, at a UCLH gym. Participants will be requested to complete two further sessions per week independently.
► Phase 2 (during hospital admission): Participants will be offered supervised exercise sessions with a physiotherapist in the hospital setting three times per week.
► Phase 3 (12 weeks after hospital discharge): Participants will be asked to continue exercising independently three times per week and will receive a weekly phone call from a physiotherapist for support and guidance.

Supervised aerobic exercise will comprise of treadmill walking or stationary cycling. Participants will be encouraged to use walking as their aerobic exercise activity, or a stationary bike or elliptical trainer if they have access to one, for independent exercise sessions. In phase 1, prior to transplant admission, aerobic exercise intensity will be targeted between 60% and 80% of heart rate (HR) reserve (or Karvonen formula), in which target HR = [(max HR – resting HR) × % intensity]+resting HR, where maximum HR=220 – age.[26] Duration will be started at 15 min and progressed by 5 min/week to achieve a minimum of 30 min by week 3–4. This will be further progressed to 40 min by week 5. Target HR will be monitored using HR belts and participants will also use a Rating of Perceived Exertion (RPE) Scale.[27] The participants will be given a scale in their exercise log book, instructed in its use and advised to work to levels of exertion as determined under supervision. Participants will be encouraged to reach the target duration in one bout but in the presence of discomfort or wish to change exercise machine/method then participants will be required to complete the target duration in bouts of 15 min or more.

Resistance exercises will be prescribed to target all major muscle groups (depending on individual health and contraindications). Resistance training will be performed using body weight, progressed with weights or using elastic resistance bands of various strengths. These bands will be supplied to participants at the start of the study to allow convenient independent resistance training. Starting resistance and repetitions of the exercises will be informed by a 10-repetition maximum assessment carried out by the physiotherapist during the first session. Individually tailoring and gradual progression in resistance training will be prescribed as deemed appropriate at each exercise session by the physiotherapist according to published principles.[28] Participants will be provided with exercise log books and asked to record what exercise they do in the books, as well as document how they feel after exercising and any reasons for not carrying out the exercise as prescribed.

In phase 2, when participants are admitted to hospital for their ASCT, they will be offered supervised exercise with the physiotherapist three times per week in the ambulatory care or hospital ward setting (depending on each participant's clinical location), which they can decline if they wish to do so. It is anticipated that participants will be symptomatic from the effects of their ASCT, which may include nausea, gastrointestinal complaints and fatigue. They may also experience bleeding and infection as a result of thrombocytopenia and low white cell counts. Therefore the exercise intervention will be highly individualised and tailored according to participant's capacity to carry out the exercises. Aerobic exercise will be carried out, as able, on a stationary bike in bouts of 10 min, up to a total duration of 30 min. HR will be monitored throughout and participants will be guided to exercise at a RPE (determined by RPE scale) they are comfortable with. Any exercise undertaken, declined or not possible due to clinical indication will be recorded by the physiotherapist in the participants' log book.

In phase 3, when the participants have been discharged from hospital they will be advised by the physiotherapist to continue the independent exercise programme, tailored to their fitness on discharge, until they are 12 weeks post-ASCT. During these 12 weeks, they will be contacted once per week via telephone by the physiotherapist to provide support and guidance with continuing to exercise regularly during their recovery and to assist them in progressing their exercise programme. The participants will be encouraged to meet recommended guidance for levels of PA[29] and requested to record any exercise undertaken in their log books.

Previous systematic reviews and meta-analyses have shown that BCTs from exercise interventions that are associated with improved adherence to exercise by cancer survivors include 'goal setting (behaviour)', setting 'graded tasks', and 'instruction of how to perform behaviour'[30 31] practice and self-monitoring and encouraging participants to attempt to generalise exercise behaviours learnt in supervised exercise environments to other, non-supervised contexts.[32] According to the BCT Taxonomy v1, strategies to promote adherence to the intervention and promote change in exercise behaviour in the current intervention have been coded.[33] See table 1 for incorporated BCTs and examples of how they have been included in the intervention. Physiotherapists who deliver the intervention will receive training, clinical supervision and will be provided with a standard operating procedure to carry out the intervention and support the participants to exercise and incorporate BCTs.

### Usual care (control) group

Participants randomised to usual care will receive the usual advice provided by haematology CNSs. Participants who specifically ask the research team for exercise advice will be signposted to generic PA advice offered for people undergoing cancer treatment on the Macmillan Cancer support website[34] or from the information and support service within UCLH.

During hospital admission, patients undergoing SCT are routinely screened by a hospital physiotherapist and receive input for any functional or mobility-related deficits that may prevent or delay hospital discharge. All patients within the study who require physiotherapy or occupational therapy during their hospital admission will receive input as indicated. The indication for and details of input required will be obtained from participants' hospital records and recorded in the study file for enrolled participants who require additional therapy input. Control group participants will not be asked to monitor their activity or receive log books.

### Outcome assessment

The primary end points of this pilot study will be study feasibility. A target recruitment rate of >50% of potential participants screened as eligible and approached for this

**Table 1** BCT coded to BCT taxonomy (BCTT V1) Michie et al, 2013[33]

| BCT label | BCT no. (BCTT v1) | Component of intervention |
|---|---|---|
| Goal setting (behaviour) | 1.1 | Overarching goal of exercise programme for all participants is to exercise three times per week. Participants also supported to define own personal subgoals, which are recorded in log book. |
| Problem-solving | 1.2 | Enablers, barriers and solutions to barriers completed with physiotherapist and recorded in log book. Problem-solving with physiotherapist if discrepancy between current behaviour and goal is identified. |
| Action planning | 1.4 | Attendance at supervised exercise session in gym planned for specific day each week. Participants are also supported to plan independent exercise sessions (eg, time/day of the week) with physiotherapist. These plans are recorded in log book prior to planned execution. |
| Review behaviour goal | 1.5 | Participants review performance of exercise sessions recorded in log book against planned goals and consider modifying goals accordingly with physiotherapist in supervised exercise sessions. |
| Discrepancy between current behaviour and goal | 1.6 | Physiotherapist informs participant if there is a discrepancy between current behaviour and goal. |
| Feedback on behaviour | 2.2 | Feedback on performance of exercise programme delivered by physiotherapist during supervised exercise sessions (phases 1 and 2) and telephone calls (phase 3). |
| Self-monitoring of behaviour | 2.3 | Participants asked to record exercise carried out each week in log book. Heart rate monitoring used in supervised and independent exercise sessions to monitor behaviour being carried out at target intensity. |
| Biofeedback | 2.6 | Heart rate monitoring of aerobic exercise effort. |
| Instruction on how to perform the behaviour | 4.1 | Supervised exercise sessions with physiotherapist who instructs/teaches participant how to perform behaviour. |
| Information about health consequences | 5.1 | Physiotherapist and log book provide information about effects of exercise for health in context of myeloma. |
| Monitoring of emotional consequences | 5.4 | Participants encouraged to complete weekly reflections in log book. |
| Information about emotional consequences | 5.6 | Physiotherapist and log book provide information about effects of exercise on emotional well-being. |
| Demonstration of the behaviour | 6.1 | Physiotherapist demonstration and teaching of exercise programme in supervised sessions. |
| Behavioural practice/rehearsal | 8.1 | Exercises taught/demonstrated in supervised sessions are practiced within those sessions and independent sessions (phases 1 and 2). |
| Generalisation of target behaviour | 8.6 | Advice to perform exercise programme, learnt in supervised session, in independent sessions, during hospital admission and post-discharge. |
| Graded tasks | 8.7 | Progression of exercise sessions throughout exercise period. |
| Credible source | 9.1 | Education and prompting from a physiotherapist with expertise in myeloma/haematology and provided in log book. |
| Pros and cons | 9.2 | Prompts to consider advantages and disadvantages of exercising through discussion with physiotherapist and recording in log book. |
| Adding objects to the environment | 12.5 | Provision of heart rate monitors, resistance exercise bands and exercise programme sheets/log book for use in independent sessions. |
| Verbal persuasion about capability | 15.1 | Verbal support and supervision of physiotherapist to encourage exercise sessions before, during and after transplant treatment. |

BCT, behaviour change techniques.

study will be a primary indication of feasibility. Feasibility of delivering the intervention protocol will be assessed by evaluating adherence to the intervention and acceptability of receiving the intervention (including deviations from exercise protocol). Attrition due to the intervention, loss to follow-up and adverse events will also be assessed.

A number of secondary end points will also evaluate any between group differences and determine intended primary and secondary end points for a fully powered trial. Fatigue will be assessed using the Functional Assessment of Chronic Illness Therapy (FACIT-F) questionnaire.[35] QOL will be measured with two complementary instruments: the European Organisation for Research and Treatment of Cancer QOL Questionnaire (EORTC QLQ-C30 and myeloma-specific module QLQMY20)[36 37] and the Functional Assessment of Cancer Therapy Bone Marrow Transplantation (FACT-BMT) scale questionnaire.[38] The EORTC QLQ C-30 has a focus on physical and functioning dimensions of QOL whereas FACT-BMT can assess impact of treatment and side effects on QOL. Comparison analysis indicates that both instruments should be used together rather than in place of each other when assessing QOL in patients undergoing SCT.[39 40]

Functional capacity will be assessed with a 6 min walk test (6MWT), handheld dynamometry and a timed sit-to-stand test. PA and sedentary time will be recorded over 5–7 days using an ActivPAL accelerometer (PAL Technologies, Glasgow, UK). Additional measures of self-reported PA behaviour will be captured using the self-complete short form International Physical Activity Questionnaire[41] and the Exercise Self-Efficacy Scale.[42] All these measures will be recorded at baseline and three time points during the ASCT pathway. At the final time point, an adapted Client Service Receipt Inventory questionnaire will be used to assess health and social care usage in the time following hospital discharge.[43]

The outcomes measures and time points for data collection are summarised in table 2.

## Qualitative data

Semi-structured interviews will be conducted with a purposeful sample of participants at two time points. Study 'decliners', deemed eligible for the study, who have received a study PIS but declined to enrol in the study will be asked if they would participate in a short interview. These interviews will take place either face to face in the clinic setting or via telephone (based on participants' preference) and will be conducted to explore the patients' experiences of being approached for the exercise study and the process and outcome of their decision-making process in relation to the exercise study.

Additional semi-structured interviews from approximately 20 participants who enrol and take part in the study will take place at final follow-up, approximately 3 months post-transplant, and will be conducted to explore the participants experiences of ASCT treatment and their experiences of being enrolled in the study.

Participants will be purposively sampled to include participants from both the intervention arm and the control arm.

## Anticipated dates of trial commencement and completion

Recruitment commenced in June 2019, with data collection for all follow-up assessments estimated to be completed by December 2020.

## Statistical analysis including sample size calculations

This is a pilot study with no formal power calculation. There is limited available data on the application of an exercise intervention delivered prior to ASCT in people with myeloma. This study will provide estimates of means/SD for the intended outcomes of a fully powered RCT, to support a sample size calculation for a larger trial. Meaningful clinically important differences are known for a number of the secondary end points, including the FACIT-F,[44] EORTC-QLQ-30 and MY20 module[45 46] and 6MWT[47] and will be used to inform future sample size calculations. For a pilot study, including >30 participants in each arm is considered to be an acceptable number.[48 49]

## Data analysis

The primary outcomes (intervention recruitment rate and intervention adherence) from the trial will be summarised as percentages or mean values for each item, with 95% CIs. Data on the acceptability of the randomisation through its effect on dropout rates, completion rates of the outcome assessments and reported use and satisfaction with the intervention materials will also be analysed descriptively. Group differences in fatigue, QOL, functional capacity, PA and sedentary time, self-efficacy at each time point will be analysed using linear regression for continuous factors (eg, fatigue) and frequency tables, $\chi^2$ tests and logistic regression for categorical factors. Gender and age will be included within the models as covariates. Repeated measures analyses (eg, mixed models) will be used to analyse the baseline and follow-up scores of all measures together. Assessments for each outcome will be made to determine whether the data are normally distributed. For outcomes that are not, even after appropriate transformations, non-parametric methods will be used for data analyses at specific time points.

Qualitative data derived from the interviews of patients will be analysed thematically using an approach that is both deductive and inductive to ensure that the full range of participants' responses are represented. Codes generated from the data will be assigned to portions of the text to develop an initial coding framework, which will be used by another researcher familiar with the sample population to independently double-code a proportion of the interview transcripts. The framework will be revised to ensure that the codes accurately reflect the data and any discrepancies will be resolved via discussion. The final codes will be applied to the data and incorporated into appropriate themes/subthemes.

**Table 2** Study schedule of assessments

| Visit no. | Screening | Baseline assessment (T0) | Follow-up 1 (T1) | Follow-up 2 (T2) | Follow-up 3 (T3) |
|---|---|---|---|---|---|
| | | 1 | 2 | 3 | 4 |
| | On referral to UCLH ASCT MDT | On day of first transplant clinic appointment | Day of transplant | Day of discharge from hospital | Post-transplant follow-up clinic at 3 months |
| Qualitative interview (~20 patients who decline study) | X | | | | |
| Window of flexibility for timing of visits | | ±2 days | 0±1 day | 0±1 day | ±5 days |
| Screening eligibility | X | | | | |
| Informed consent | | X | | | |
| Demographics and clinical history | | X | | | |
| Fatigue: FACT-F | | X | X | X | X |
| QOL: EORTC QLQ C-30-MY20 | | X | X | X | X |
| QOL: FACT-BMT | | X | X | X | X |
| Exercise behaviour: IPAQ-SF | | X | X | X | X |
| Exercise Self-efficacy Scale | | X | X | X | X |
| Functional capacity: 6MWT | | X | X | X | X |
| Functional capacity: 30 s STS | | X | X | X | X |
| Functional capacity: handheld dynamometry | | X | X | X | X |
| Resting BP and HR | | X | X | X | X |
| Height and weight | | X | X | X | X |
| Accelerometery (3–7 days) | | X | X | X | X |
| Qualitative interview (20 participants) | | | | | X |
| Health and social care service use: CRSI | | | | | X |
| Blood counts | | X | X | X | X |
| Levels of immune function (blood) | | X | X | X | X |

ASCT, autologous stem cell transplant; BP, blood pressure; EORTC QLQ C-30-MY20, European Organisation for Research and Treatment of Cancer QOL Questionnaire and myeloma specific module; CRSI, Client Service Receipt Inventory; FACT-BMT, Functional Assessment of Cancer Therapy Bone Marrow Transplantation ; FACT-F, Functional Assessment of Chronic Illness Therapy Fatigue questionnaire; HR, heart rate; IPAQ-SF, International Physical Activity Questionnaire short form; MDT, multidisciplinary team; 6MWT, 6 min walk test; QOL, quality of life; STS, sit to stand; UCLH, University College London Hospitals NHS Foundation Trust.

## ETHICS AND DISSEMINATION
All participants will provide informed written consent to take part and can withdraw at any time.

Results of this study will be written up as part of a doctoral thesis and disseminated through publication in peer-reviewed journals, presentation at haematology and rehabilitation-related meetings and shared locally. A lay summary of the study results will be provided to participants who request to receive one after the study has completed.

## DISCUSSION
The study described in this protocol will evaluate the feasibility of carrying out an exercise intervention among people with myeloma before, during and after ASCT at a UK centre. There is no standardised or accepted approach to using lead time before SCT to prepare and physically optimise patients and the importance of carrying out research of this kind is particularly pertinent in this population for whom SCT currently offers the best possibility of longer progression-free survival, yet carries the risk of considerable morbidity.

The strengths of this study include the use of a pragmatic intervention using BCTs and designed to integrate into the existing ASCT treatment pathway at our centre and the use of a partly supervised exercise intervention among myeloma patients preparing for transplant. This study is also the first of its kind in the UK to incorporate a control arm. If the feasibility criteria for trial is met and this single-site, pilot trial is deemed to be successful, it

is anticipated to be an easily replicable trial design for future evaluation as a fully powered multicentre trial. Recent guidance for prehabilitation in cancer advocates for a multimodal intervention, comprising of exercise, nutrition and psychological/behaviour change assessment and support.[50] There is no nutritional assessment or dietary intervention included in this study which may be considered a limitation of this exercise-focused intervention. It is expected that this pilot study will be acceptable and feasible as an important step in developing the field of exercise prehabilitation and rehabilitation within haematological cancer treatment in the UK. Dissemination of study results is expected in early 2021.

**Author affiliations**
[1]Therapies & Rehabilitation, University College London Hospitals NHS Foundation Trust, London, UK
[2]Research Department of Haematology, Cancer Institute, University College London, London, UK
[3]Research Department of Behavoural Science and Health, University College London, London, UK
[4]Queen Square Centre for Neuromuscular Diseases, University College London, London, UK
[5]Department of Haematology, University College London Hospitals NHS Foundation Trust, London, UK

**Contributors** OM drafted the manuscript. OM, KY, AF and GR designed the study and secured funding and ethical approval. ALR, JL and NR provided intellectual input to the final study protocol according to their area of expertise. All authors contributed to revising and final approval of the submitted manuscript.

**Funding** This report is an independent research supported by the National Institute for Health Research (HEE/ NIHR ICA Programme Clinical Doctoral Research Fellowship, Ms Orla McCourt, ICA-CDRF-2017-03-067).

**Disclaimer** The views expressed in this publication are those of the author(s) and not necessarily those of the NHS, the National Institute for Health Research or the Department of Health and Social Care.

**Competing interests** None declared.

**Patient consent for publication** Not required.

**Ethics approval** This study has been approved by the NHS Health Authority Research Ethics Committee—Camden and Kings Cross—reference 19/LO/0204 (Protocol Version 1.0, 14 December 2018).

**Provenance and peer review** Not commissioned; externally peer reviewed.

**ORCID iD**
Orla McCourt http://orcid.org/0000-0001-7572-2540

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
