## [Reviewer comments · BMJ Open]

ARTICLE DETAILS

TITLE (PROVISIONAL)	PERCEPT myeloma: a protocol for a pilot randomised controlled trial of exercise prehabilitation before and during autologous stem cell transplantation in patients with multiple myeloma
AUTHORS	McCourt, Orla; Fisher, Abigail; Ramdharry, Gita; Roberts, Anna; Land, Joanne; Rabin, Neil; Yong, Kwee

VERSION 1 – REVIEW

REVIEWER	Bart Staal Radboud university medical centre Nijmegen. HAN university of applied sciences Nijmegen The Netherlands
REVIEW RETURNED	07-Sep-2019

GENERAL COMMENTS	The authors are to be congratulated with this well designed study. I only have a few comments. 1. In the abstracts the primary outcomes (feasibility measures) are lacking2. the timing of the first measurement (6-8 weeks prior to SCT) could be reported as well in the abstract.3. The exercise training in phase 1 will be conducted in a gym at UCL. Is it feasible for study participants to visit UCL regularly in this period (don't know there travel distances) or did you consider outpatient physiotherapy interventions more close to their home address.4. To what extent will study physiotherapists be trained to deliver the intervention. I mean behavioral change techniques are not always easy to apply.5. Descriptive statistics of this study will be used to support a future sample size calculation. These effect estimates do not necessarily be clinically important. Do you have any idea what could be a clinically important effect size for cancer-related fatigue in this population?
--

REVIEWER	Shinichiro Morishita Niigata University of Health and Welfare, Japan
REVIEW RETURNED	11-Oct-2019

GENERAL COMMENTS	Author investigates the feasibility of a physiotherapist-led exercise intervention as an integral part of the myeloma autologous SCT pathway at a UK tertiary centre. However, I think a great deal of the material covered in the current paper has been addressed in a lot of previous researches. A lot of studies have investigated that the effect of physical exercise in patients with allo-HSCT. Thus, I think this RCT study is unlikely to be of interest to readers.
---

REVIEWER	ILKE KESER GAZI UNIVERSITY, FACULTY OF HEALTH SCIENCES DEPARTMENT OF PHYSIOTHERAPY AND REHABILITATION
REVIEW RETURNED	14-Nov-2019

GENERAL COMMENTS	The manuscript is about a protocol produced by physiotherapy team, as a part of a doctorate thesis, which is an important point in the rehabilitation of MM. Although the manuscript is well written, but the results part is not included, data analyses were written as will be applied. Without results, and so short discussion without reference the manuscript is not ready to publication. there are some offers to improve understanding: 1- Title: Multiple is needed to be added before myeloma as it is written in literature. 2- Abstract: In method part, -the groups is needed to be identified and mentioned separately. -sometimes the name of tests sometimes the topic of the test is written. There is a need to standardize the writing. Key words, is needed to be justified to whole line Strengths and limitations are needed to be justified to whole line 3- Introduction Multiple is needed to be added before myeloma as it is written in literature. MM can be used as abbreviation. Page 5, 2nd paragraph "intervention" is repeated several times there is a need reorganize Objectives can be shortened 4- Method Page 7. CNS is need to be explained? It is mentioned that there is a rehabilitation team decided to assess or treatment Page 8 Inclusion and exclusion criteria can be written in the text, table is not usual for this aim. It is familiar to write the criteria in the text. Page 9 "There is no plan to replace..." sentence can be omitted. Page 10 "the study physiotherapist" term is needed to be controlled. Just "physiotherapist" is enough Page 12 comma before (25,26) can be deleted the numbers of the tables are needed to be checked and corrected. the abbreviations are needed to be written below the table and explained their meanings. Page 13 "Heart rate" can be used as HR, The numbers in the table can not be understood. 1.6 5.4...??? Page 14 "Heart rate" can be used as HR, Page 16 the numbers of the tables are needed to be checked and corrected. the abbreviations are needed to be written below the table and explained their meanings. Page 17 ":" and the letters after this sign is needed to be checked if it is really needed Page 18 the abbreviations are needed to be written below the table and explained their meanings. Because the table took place more than one page the meaning of the columns can be written in last part of the table in the page 18. "Anticipated dates..." this information is necessary. Can it be omitted? Page 20 Ethics part is generally in the methods. Please check if it can be transferred to method part. "Results..." paragraph can be written below the manuscript as an additional information. 5- Results
--

	The results part can not be seen. Although data analysis part is written but the results section is missing. The analysis results are not mentioned in the text and in tables. 6- Discussion This part is needed to be improved and supported by literature. There is no reference in this part. It is clear that there is a new protocol produced in this manuscript. But the content of the protocol has common physiotherapy approaches. So you may discuss the necessity, importance, benefits of this protocol in comparison to other methods. 7- References Reference 7,10,19,20,25,26,27,30 are needed to be controlled as writing rules of the journal.
--	--

VERSION 1 – AUTHOR RESPONSE

Reviewer: 1

The authors are to be congratulated with this well designed study. I only have a few comments.

Response: Thank you for your positive comments on our manuscript and your helpful revisions.

1. In the abstracts the primary outcomes (feasibility measures) are lacking.

Response: We thank the reviewer for pointing this out, and have now added the feasibility measures to the abstract.

2. the timing of the first measurement (6-8 weeks prior to SCT) could be reported as well in the abstract

Response: This has now been added to the abstract. The abstract has been edited to accommodate the extra detail within the maximum word allowance.

3. The exercise training in phase 1 will be conducted in a gym at UCL. Is it feasible for study participants to visit UCL regularly in this period (don't know there travel distances) or did you consider outpatient physiotherapy interventions more close to their home address.

Response: We thank the reviewer for this important comment. Due to the sequelae of osteolytic bone destruction in these patients (vertebral and long bone fractures and deconditioning), we designed the study to incorporate review by a specialist physiotherapist with experience of working with myeloma patients at our centre, to achieve appropriate tailoring of programme to individual needs and fitness, and for standardisation of the pilot intervention. Once the safety of our methods has been established in this clinical environment, expanding the intervention to local communities would be a goal for a larger national trial. We have also added a sentence in the introduction to highlight the challenge of bone disease limiting physical activity in this patient group (page 4, line 68-72).

4. To what extent will study physiotherapists be trained to deliver the intervention. I mean behavioral change techniques are not always easy to apply.

Response: We thank the reviewer for this important point. All research physiotherapists receive training on intervention delivery and behavioural change techniques. We have added this to the manuscript, as well as a Table detailing behaviour change techniques included in the intervention (Page 14-15).

5. Descriptive statistics of this study will be used to support a future sample size calculation. These effect estimates do not necessarily be clinically important. Do you have any idea what could be a clinically important effect size for cancer-related fatigue in this population?

Response: We thank the reviewer for raising this pertinent point. With reference to the particular tools that we use, clinically important change in QOL, fatigue and 6MWT are now referred to in the

statistical analysis section (Page 19, line 333-335).

Reviewer 2

Author investigates the feasibility of a physiotherapist-led exercise intervention as an integral part of the myeloma autologous SCT pathway at a UK tertiary centre. However, I think a great deal of the material covered in the current paper has been addressed in a lot of previous researches. A lot of studies have investigated that the effect of physical exercise in patients with allo-HSCT. Thus, I think this RCT study is unlikely to be of interest to readers.

Response: We appreciate the opportunity to clarify the current lack of evidence of exercise training specifically in multiple myeloma patients undergoing autologous stem cell transplantation. We have referred to the literature on exercise in patients undergoing transplantation, including two systematic reviews which highlight issues with the heterogeneity of the study samples (Page 5, line 73). These reviews of 8 and 11 studies respectively include only 2 studies focussed on a homogeneous autologous stem cell transplant study population. Those two studies remain the only published RCTs conducted in autologous myeloma populations (Coleman et al, 2003; Coleman et al, 2008) and are referenced within our introduction (Page 5, line 90-91). We have included a critique of those studies and a justification for further studies among people with myeloma preparing for autologous transplantation.

We agree with the reviewer that there is a greater amount of research detailing the role of exercise among allogeneic stem cell transplant recipients, summarised in a review of the benefits among this population, which he has recently co-authored (Morishita, et al, 2019). Studies among allogeneic transplant recipients highlight the intensive nature of the medical intervention and its associated symptom burden and consequences, most notably graft versus host disease (GvHD) and an increased risk of treatment associated mortality. In contrast, with autologous transplantation the mortality is considerably lower and there is no risk of GvHD, with its attendant risk of infections and short and long term morbidity. Previous studies among allogeneic, autologous and mixed auto and allo-transplant recipients have indicated promising effects in health-related quality of life and a range of functional capacity endpoints. Our study on a uniform patient population will contribute to the limited evidence in myeloma patients for exercise before and during first-line treatment. This study will add to our understanding of the safety and feasibility of an exercise intervention among myeloma patients, who, although undergoing a less intensive procedure than an allotransplant and unaffected by GvHD, are generally older patients, many of whom are suffering from the sequelae of myeloma-related bone disease. The higher population age, inclusion of patients with evidence of bone disease at diagnosis and the lack of transplant exercise studies conducted in the UK, we feel will make this manuscript interesting to both academic and clinical readers.

We have added two additional sentences in the Introduction to help address the points raised. Firstly, we have added a citation to highlight the need to develop exercise intervention evidence in the presence of myeloma related bone disease, page 4, line 68. Secondly, as we had not specified the lack of autologous only exercise studies included in the previously cited reviews, we have added a sentence to our introduction to highlight this (Page 5, line 79-89).

Reviewer: 3

The manuscript is about a protocol produced by physiotherapy team, as a part of a doctorate thesis, which is an important point in the rehabilitation of MM.

Although the manuscript is well written, but the results part is not included, data analyses were written as will be applied. Without results, and so short discussion without reference the manuscript is not ready to publication.

Response: We thank the reviewer for raising this point. We have written our manuscript as a protocol paper, outlining a study in progress, hence there are no results to publish in this manuscript. Our intention by publishing this study protocol is to communicate ongoing research in this understudied area and - along with registration of the trial - to promote transparency of methods when reporting the

future study results. The discussion highlights a brief summary of the ongoing project, its potential strengths and limitations.

there are some offers to improve understanding:

Response: we thank the reviewer for all these helpful suggestions and can confirm that we have implemented these changes in our amended manuscript.

1- Title: Multiple is needed to be added before myeloma as it is written in literature.

Response: We have now added the word 'multiple' to the title.

2- Abstract:

In method part,

-the groups is needed to be identified and mentioned separately

-sometimes the name of tests sometimes the topic of the test is written. There is a need to standardize the writing.

Key words, is needed to be justified to whole line

Strengths and limitations are needed to be justified to whole line

Response: We have added the groups and addressed these formatting issues.

3- Introduction

Multiple is needed to be added before myeloma as it is written in literature. MM can be used as abbreviation.

Response: As well as adding 'multiple' to the title, we have added 'also known as multiple myeloma' added to introduction (Page 4, line 46).

Page 5, 2nd paragraph "intervention" is repeated several times there is a need reorganize

Response: We thank the reviewer for this observation. We have removed and/or replaced 'intervention' to clarify the paragraph on page 5-6, between lines 90-102.

Objectives can be shortened

4- Method

Page 7. CNS is need to be explained? It is mentioned that there is a rehabilitation team decided to assess or treatment

Response: Clinical Nurse Specialist role and clinical MDT description clarified on page 7, line 136 – this is the medical and nursing team not a rehabilitation team.

Page 8 Inclusion and exclusion criteria can be written in the text, table is not usual for this aim. It is familiar to write the criteria in the text.

Response: We thank the reviewer for this comment. We have removed the inclusion/exclusion table and added the criteria as text in the methods section, on page 7. Other table numbers updated accordingly.

Page 9 "There is no plan to replace..." sentence can be omitted.

Response: Thank you for this suggestion. We have removed this sentence.

Page 10 "the study physiotherapist" term is needed to be controlled. Just "physiotherapist" is enough

Response: We have removed 'study' where indicated.

Page 12 comma before (25,26) can be deleted

Response: we have deleted this comma.

the numbers of the tables are needed to be checked and corrected. the abbreviations are needed to

be written below the table and explained their meanings

Response: We have updated the table numbers and added abbreviations below the tables.

Page 13 “Heart rate” can be used as HR,

The numbers in the table can not be understood. 1.6 5.4...???

Response: We appreciate the observation of the reviewer and have edited the table to provide clarity.

These numbers correspond to the specific descriptors within the published Behaviour Change Taxonomy, which advocates standard reporting for use of behaviour change techniques in intervention description and reporting. We have changed the table accordingly to clarify the use of these numbers. The numbers have been moved to separate column and the column labelled. Abbreviation descriptions have also been added to table. We hope this clarifies the numbers in the table.

Page 14 “Heart rate” can be used as HR,

Page 16 the numbers of the tables are needed to be checked and corrected. the abbreviations are needed to be written below the table and explained their meanings.

Response: We have added a list of abbreviations to the table legend. In addressing this comment, we noticed that an outcome in the table had not been described within the main body of the manuscript, we have therefore added detail of the client service receipt inventory on page 17, line 306-308.

Page 17 “:” and the letters after this sign is needed to be checked if it is really needed

Response: We believe the reviewer is referring to the abbreviations on the study schedule of assessments. We have added abbreviations in full in the table legend and have kept the abbreviations in the titles to give a summary of the measures used for each study outcome.

Page 18 the abbreviations are needed to be written below the table and explained their meanings.

Because the table took place more than one page the meaning of the columns can be written in last part of the table in the page 18

Response: We thank the reviewer for these comments regarding the table. We have formatted the table to fit on one page to prevent it spreading across multiple pages and for ease of review. It is now on page 18 of the tracked manuscript. The abbreviations have been addressed as previously detailed.

“Anticipated dates...” this information is necessary. Can it be omitted?

Response: Thank you to the reviewer for this suggestion. We would like to leave the anticipated trial completion dates within the manuscript, as evident in other published BMJ Open protocols papers. We also feel that readers may wish to anticipate dissemination of results of this under researched field.

Page 20 Ethics part is generally in the methods. Please check if it can be transferred to method part.

Response: We thank the reviewer for this suggestion. We have consulted the BMJ Open author submission guidelines for protocols and ‘Ethics and dissemination’ is listed as a separate section after Methods and analysis. We have therefore kept this section separate to the methods part.

“Results...” paragraph can be written below the manuscript as an additional information.

5- Results

The results part can not be seen. Although data analysis part is written but the results section is missing. The analysis results are not mentioned in the text and in tables.

Response: We appreciate the reviewer highlighting that we have detailed an analysis plan but no results in the manuscript. As addressed in their earlier comments this manuscript describes a protocol of a study in progress. As researchers, we are publishing our study methods and a priori analysis to enhance transparency of our future research reporting. We have reviewed the BMJ Open author submission guidelines and confirmed that ‘Results’ is not a heading required in protocol manuscripts.

There are therefore no results to detail in the current manuscript.

6- Discussion

This part is needed to be improved and supported by literature. There is no reference in this part. It is clear that there is a new protocol produced in this manuscript. But the content of the protocol has common physiotherapy approaches. So you may discuss the necessity, importance, benefits of this protocol in comparison to other methods.

Response: Thank you for this comment. In addition to our response above, our discussion summarises the protocol as outlined and we have summarised the supporting literature in the introduction to the study (page 5-6, line 73-102). We agree that the intervention includes approaches commonly applied within physiotherapy, however these approaches are not commonly applied within the study population, for reasons that we have described in the manuscript and the responses to reviewers above. This study will provide important evidence regarding the feasibility of the intervention in this group, during existing medical treatment and importantly, allow us to ascertain safety as well as efficacy as a priority. We believe this will be of interest to both clinicians and researchers.

7- References

Reference 7,10,19,20,25,26,27,30 are needed to be controlled as writing rules of the journal.

Response: We have updated the reference list according to the author guidelines.

VERSION 2 – REVIEW

REVIEWER	Bart Staal Radboud university medical centre Nijmegen
REVIEW RETURNED	20-Dec-2019
GENERAL COMMENTS	The authors are to be congratulated with this important study protocol.